# Remaining Useful Life Prediction of Gear Pump Based on Deep Sparse Autoencoders and Multilayer Bidirectional Long–Short–Term Memory Network

**Peiyao Zhang** [1,2]**, Wanlu Jiang** [1,2,*]**, Xiaodong Shi** [1,2] **and Shuqing Zhang** [3]

1 Hebei Provincial Key Laboratory of Heavy Machinery Fluid Power Transmission and Control, Yanshan University, Qinhuangdao 066004, China
2 Key Laboratory of Advanced Forging & Stamping Technology and Science, Yanshan University, Qinhuangdao 066004, China
3 Institute of Electrical Engineering, Yanshan University, Qinhuangdao 066004, China
* Correspondence: wljiang@ysu.edu.cn

**Abstract:** Prediction of remaining useful life is crucial for mechanical equipment operation and maintenance. It ensures safe equipment operation, reduces maintenance costs and economic losses, and promotes development. Most of the remaining useful life prediction studies focus on bearings, gearboxes, and engines; however, research on hydraulic pumps remains limited. This study focuses on gear pumps that are commonly used in the hydraulic field and develops a practical method of predicting remaining useful life. The deep sparse autoencoder is used to extract multi–dimensional features. Subsequently, the feature vectors are inputted to the support vector data description to calculate the machine degradation degree at the corresponding time and obtain the health indicator curve of the machine's life cycle. In building the health state degradation curve, data are processed in an unsupervised manner to avoid the influence of artificial feature selection on the test. The method is validated on the public bearing and self–collected gear pump datasets. The results are better than those of the comparative algorithms: (1) commonly used time–frequency characteristics with principal component analysis and (2) deep sparse autoencoder with self–organizing mapping. Next, the multilayer bidirectional long–short–term memory network is trained as a prediction model using the gear pump health indicator curves obtained previously and applied to the test data. Finally, the proposed method is compared with two others of the same type and the evaluation indexes are calculated based on the prediction results of the three algorithms. From the evaluation indexes, the mean absolute error of the proposed method is reduced by 2.53, and the normalized mean squared error is reduced by 0.36. This result indicates that the prediction results of the method for the remaining useful life of the gear pump are closer to the actual situation.

**Keywords:** deep sparse autoencoder; support vector data description; multilayer bidirectional long–short–term memory network; remaining useful life; gear pump



## 1. Introduction

As the complexity of mechanical equipment continues to increase, and the equipment's maintenance cost gradually rises, researchers have to find new methods to improve the reliability or predict the remaining life of the equipment [1,2]. Hydraulic systems offer significant advantages, including a large power–to–mass ratio, flexible control, and fast response speed. They are widely used in industrial machinery, mobile machinery, aerospace applications, and other fields. A complete hydraulic system typically comprises five parts—power, executive, control, and auxiliary elements, and working medium [3,4]. The power element converts mechanical energy into pressure energy [5,6]. This element is called a hydraulic pump. Hydraulic pumps typically include gear, vane, and plunger

pumps [7,8]. Among these, the gear pump is the most commonly used pump component. This study considers the gear pump as the research object and develops a prediction algorithm for its remaining useful life (RUL) [9,10].

In the 1970s, the U.S. military successfully applied the built–in testing technology to monitor the A–7E aircraft engine [11] and improve aircraft safety and reduce equipment maintenance costs. In the 1990s, the prognostics and health management (PHM) technology was applied to the engine of the F22, enabling the aircraft to have the most advanced self–checking and self–diagnosis system at that time. At the beginning of this century, the U.S. military extended the PHM technology to frame, electromechanical, transmission, and control systems and airborne electronic equipment on F35 fighters. This significantly improved the aircraft's attendance rate, safety, and maintenance costs. The successful application of PHM technology to F35 has received considerable research attention [12].

RUL prediction technology is a key component of PHM [13]. RUL is defined as the length of time a system or component can be used normally [14]. Specifically, it predicts the interval between the current time and the time of fault occurrence based on existing degradation data. RUL prediction methods can be divided into the following three categories depending on whether physical knowledge is used—prediction methods based on physical models, prediction methods based on data-driven methods, and hybrid prediction methods [15]. According to this classification, the proposed approach in this paper is a data–driven method.

In the past years, many researchers have carried out a great deal of work on adaptive feature extraction of equipment signals, equipment health state evaluation, and RUL prediction. X. Guo et al. proposed a hierarchical adaptive deep convolution neural network and applied it to bearing fault signals [16]. Z. Wang et al. proposed an adaptive spectrum mode extraction method and applied it to bearing fault signals [17]. W. Jiang et al. extracted the features of the sound signal of an axial piston pump based on Mel–frequency cepstrum coefficients (MFCCs). Extreme learning machine was used as a classifier to diagnose faults from sound features Comparing the results with others, the conclusion shows it is more advantageous [18]. S. Tang et al. used deep learning and Bayesian optimization to diagnose hydraulic piston pumps [19]. Z. Chen et al. used a sparse autoencoder (SAE) and deep belief network to fuse multisensor features and applied them to effectively identify the machine running conditions [20]. M. Zhao et al. used deep residual networks to fuse multiple wavelet coefficients and apply them to diagnose planetary gearbox faults [21]. M. Xia et al. used convolutional neural networks to fuse multiple sensors and apply them to diagnose rotating machinery [22]. J. Ben Ali et al. predicted the RUL of bearings accurately based on Weibull distribution and artificial neural network (ANN) [23]. R. Guo et al. used the Bayesian regularized radial basis function neural network to predict the RUL of a gear pump [24]. Z. Li et al. used data dimension reduction and just–in–time learning techniques to analyze the pressure signal to predict the RUL of a hydraulic pump [25]. G. Xu et al. proposed an online fault diagnosis method based on a deep transfer convolutional neural network framework. This method was validated on two bearing data sets and one pump data set [26]. Researchers have made fruitful progress in feature extraction, feature fusion, and RUL prediction. However, most researchers have studied relatively simple structures such as bearings. Research on hydraulic components is less frequent and is dominated by fault diagnosis techniques. There is a lack of research on RUL prediction for hydraulic pumps. In this area, many problems need to be solved. In terms of data, problems such as lack of data sets, poor data availability, and data that do not truly reflect the state of the equipment make it difficult to conduct effective model training. In terms of analysis methods, how to obtain valuable data from the massive data and how to effectively reflect the degradation state of hydraulic pumps are problems that need to be solved. In terms of prediction model training, the correct selection of algorithms and the quality of training data also directly affect prediction accuracy. Therefore, in this paper, a full–life test of gear pumps is carried out to address these problems, and a valid data set is established by collecting vibration data of the whole life cycle of the pump. Through comparison, an

accurate RUL prediction algorithm is found, which has a better prediction effect compared with the traditional algorithm.

In this paper, deep sparse autoencoder (DSAE) is used as a features extraction algorithm, which can perform feature extraction work self–adaptively and reduce human interference in this work. DSAE is generated based on the autoencoder (AE). AE is one of the artificial neural networks used in semi–supervised and unsupervised learning, and its function is to learn the representation of the input information by using it as a learning target. In 1985, Ackley et al. made the first attempt to develop the AE algorithm on Boltzmann machines and evaluated its representation learning ability by altering model weights [27]. In 1987, Lecun proposed AE formally as a neural network structure [28]. In the following period, AE was used for data noise reduction and dimensionality reduction. However, it has a fatal flaw. When the dimension of the hidden layer is equal to or greater than the dimension of the input layer, AE cannot achieve the desired learning effect. To solve this problem, in 2013, Le et al. proposed SAE [29]. In 2014, Liu et al. proposed a DSAE based on SAE. It achieved unsupervised pre–training of each encoder by stacking multiple SAEs, initializing parameters to a local optimal state, and obtaining more abstract and complex features [30]. After obtaining the features of the gear pump, it is necessary to obtain the deterioration curve of the gear pumps through features fusion. In this paper, support vector data description (SVDD) was chosen [31]. This algorithm is a single–value classification method based on a support vector machine (SVM) proposed in 2014 by Tax et al. It is widely used in the fields of anomaly detection, target identification, etc., by characterizing the target dataset. With the development of ANN in recent years, this technology has been used in pattern recognition, signal processing, knowledge engineering, expert systems, optimal combination, robot control, etc., and has achieved good results. In this background, this paper selects multilayer bidirectional long and short–term memory (Bi–LSTM) network as the RUL prediction algorithm for gear pumps. The multilayer Bi–LSTM network is developed on the long– and short–term memory (LSTM) network, which solves the gradient disappearance and gradient explosion problems during the training of long sequences. Compared with normal recurrent neural networks (RNN) [32], LSTM has better performance in longer sequences [33,34]. LSTM can only predict the next moment based on the information of the previous moment. However, in some problems, the output of the current moment is not only related to the previous state but also may be related to the future state. Bi–LSTM network solves this problem [35,36]. However in this paper, the model learning ability of single–layer Bi–LSTM is not strong, as found by the experiment. For this reason, the stacking of multilayer Bi–LSTM is used to ensure that the constructed prediction model can have sufficient feature extraction ability and increase its nonlinear mapping ability.

The Section 1 of this paper is the introduction, which introduces the paper's research content, background, purpose, and structure. Section 2 is the theoretical background, which provides a detailed description and mathematical derivation of the algorithms used in this paper. The algorithms presented include DSAE, SVDD, and multilayer Bi–LSTM. Section 3 is the experiment and calculation. First, the full–life test of gear pumps is presented in this section. After that, the data obtained from the test are substituted into the algorithm presented in Section 2 to obtain the RUL prediction results, and the proposed method is compared quantitatively with two other methods. The Section 4 is the conclusions, which summarizes the whole paper, shows the advantages of the proposed method and gives an outlook on the research. Figure 1 shows the structure of this paper.

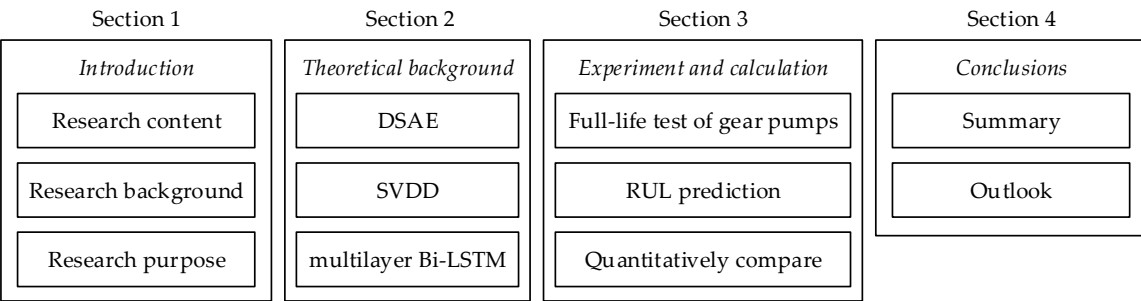

**Figure 1.** The structure of this paper.

## 2. Theoretical Background

### 2.1. DSAE

In 1987, AE was proposed as a neural network structure. In 2013, SAE was proposed based on AE that could learn useful information even when the dimension of the hidden layer was equal to or greater than that of the input layer. In 2014, Liu et al. proposed a DSAE based on SAE. It achieved unsupervised pre–training of each encoder and obtained more abstract and complex features.

#### 2.1.1. SAE

Before introducing the DSAE, it is necessary to introduce AE, which serves as its foundation. Its structure comprises an input layer, a hidden layer, and an output layer. Like the multilayer perceptron, the neurons in each AE layer are fully connected. The dimension $m$ of the hidden layer is typically smaller than the dimension $n$ of the input layer to learn important features. The gradient descent method is generally used to train and adjust the parameters. Figure 2 shows a schematic of the AE.

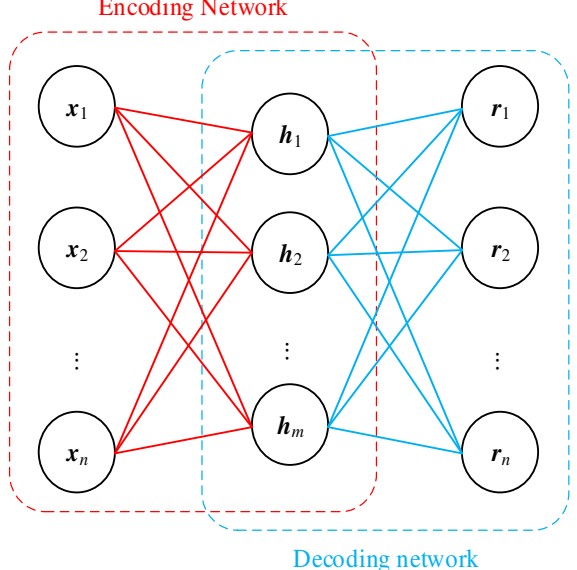

**Figure 2.** Schematic of the AE.

However, when the dimension of the hidden layer $h$ is equal to or greater than the dimension of the input layer $x$, AE cannot achieve the desired learning effect. Because of this limitation, Le et al. added sparsity constraints to the hidden layer $h$ such that the improved AE could learn effectively even when the dimension of the hidden layer $h$ was equal to or greater than the dimension of the input layer $x$. This improved AE is called the SAE.

Assuming that the data $\mathbf{X} = \{x_1, x_2, x_3, \ldots, x_i, \ldots, x_l\}$ comprising $l$ samples are input to the SAE, the number of neurons in the hidden layer of the SAE is $m$. Let $a_j$ represent the

activation value of the $j$-th neuron in the hidden layer $h$ and $a_j(x_i)$ represent the activation value of the $j$-th neuron when the input is $x_i$. The average activation value of the $j$-th neuron in the hidden layer $h$ can be expressed as

$$\hat{\rho}_j = \frac{1}{l}\sum_{i=1}^{l} a_j(x_i) \tag{1}$$

here, $\rho$ represents the sparsity parameter, and $\hat{\rho}_j$ is approximately equal to $\rho$. The sparsity parameter $\rho$ is typically close to zero (e.g., 0.05). In the SAE, the neuron activity of the hidden layer $h$ must be close to zero to meet this condition. To achieve this constraint, adding a penalty term to the optimization objective function is necessary to penalize the difference between $\rho$ and $\hat{\rho}_j$ and stabilize the average activity of neurons in the hidden layer $h$ at a low level. The penalty term is obtained according to the KL divergence and is expressed as follows [37]:

$$\sum_{j=1}^{m} KL(\rho\|\hat{\rho}_j) = \sum_{j=1}^{m}\left[\rho\log\frac{\rho}{\hat{\rho}_j} + (1-\rho)\log(\frac{1-\rho}{1-\hat{\rho}_j})\right] \tag{2}$$

The loss function of the SAE adds sparsity constraints based on AE [38], which can be expressed as

$$L_{SAE} = L(\boldsymbol{x},\boldsymbol{r}) + \beta\sum_{j=1}^{m} KL(\rho\|\hat{\rho}_j) \tag{3}$$

where $\beta$ is the penalty factor used to control the weight of the sparsity constraint. In addition, the activation function can only use the sigmoid function because the output of the hidden layer must be controlled between 0 and 1. If the tanh function is used, the calculation of the KL divergence will be incorrect when the output of the hidden layer neuron is negative.

### 2.1.2. DSAE

Deep neural networks are achieved by stacking multiple single–layer neural networks that express complex functions with fewer parameters. This approach is used in the DSAE. Figure 3 shows the DSAE network structure.

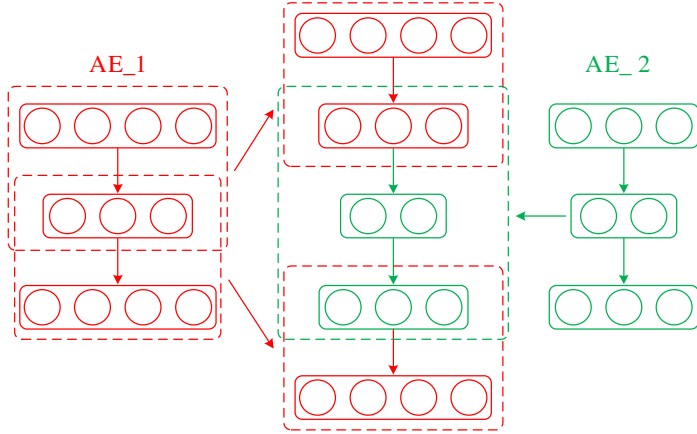

**Figure 3.** Schematic of the DSAE network. The first AE is in red, and the second AE is in green. Their position in the DSAE can also be color–coded.

In the DSAE structure, the hidden layer of each SAE is used as the input layer of the next SAE starting from the second SAE. During the stacking process, unsupervised training is applied to each stacked SAE in greedy layer–wise pre–training such that the parameters of each layer of neurons can be optimized to a local optimum state [39]. When the stacking

is complete, the network is fine–tuned to ensure that the DSAE can obtain more abstract and complex features than a single SAE.

### 2.2. SVDD

SVDD is a single–value classification method. It finds a minimum hypersphere that can include all target samples, and the samples that are outside and inside the hypersphere are considered non–target and target samples, respectively. Figure 4 shows a schematic of the SVDD in two–dimensional space.

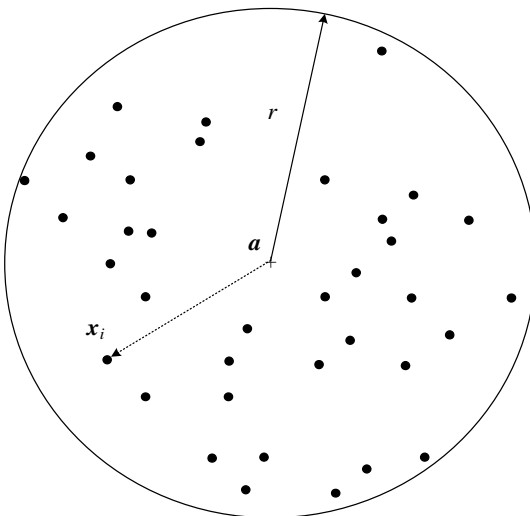

**Figure 4.** Schematic of SVDD in two–dimensional space.

In Figure 4, $a$, $r$, and $x_i$ represent the center of the hypersphere, the radius of the hypersphere, and the $i$–th data sample, respectively.

The training samples are denoted by $D = \left\{ (x_i, y_i) \middle| x_i \in R^{d \times N} \right\}$, and all of them are considered as target samples. Here, $y_i$ is the sample label set as 1, $d$ is the feature dimension, and $N$ is the number of samples. Thus, the optimization problem is expressed as follows:

$$\min_{r, a, \xi} r^2 + P \sum_{i=1}^{N} \xi_i$$
$$\text{s.t. } \|x_i - a\|^2 \leq r^2 + \xi_i, \ i = 1, 2, \cdots, N \tag{4}$$

In Equation (5), $P$ is used as a penalty factor to control the hypersphere size and sample misclassification rate. $\xi_i$ ($\geq 0$) is a slack variable. Its significance is to avoid the influence of alternative outliers in the training set and improve the generalization ability of the model. Moreover, Lagrangian factors $\alpha_i$ and $\beta_i$ are equal to or greater than zero. The original optimization problem becomes

$$L(r, a, \alpha_i, \beta_i, \xi_i) = r^2 + P \sum_{i=1}^{N} \xi_i - \sum_{i=1}^{N} \alpha_i \{ r^2 + \xi_i - (\|x_i\|^2 - 2a \cdot x_i + \|a\|^2) \} - \sum_{i=1}^{N} \beta_i \xi_i \tag{5}$$

Setting the partial derivatives of $L$ to $r$ and $a$ and $\xi_i$ to zero results in the following constraints:

$$\begin{cases} \frac{\partial L}{\partial r} = 0 \rightarrow \sum_{i=0}^{N} \alpha_i = 1 \\ \frac{\partial L}{\partial a} = 0 \rightarrow a = \sum_{i=1}^{N} \alpha_i x_i \\ \frac{\partial L}{\partial \xi_i} = 0 \rightarrow P - \alpha_i - \beta_i = 0 \end{cases} \tag{6}$$

Substituting Equation (7) to Equation (6) yields the final optimization objective, expressed as follows:

$$\max \sum_{i=1}^{N} \alpha_i \mathbf{x}_i \mathbf{x}_i^T - \sum_{i=1}^{N} \sum_{j=1}^{N} \alpha_i \alpha_j \mathbf{x}_i \mathbf{x}_j^T$$

$$\text{s.t.} \sum_{i=1}^{N} \alpha_i = 1, 0 \le \alpha_i \le P \tag{7}$$

The hypersphere center $\mathbf{a}$ can be solved. Its decision function is

$$f(x) = \text{sign}(r^2 - \|\mathbf{x} - \mathbf{a}\|^2) \tag{8}$$

The hypersphere radius $r$ is

$$r^2 = \|\mathbf{x}_{sv} - \mathbf{a}\|^2 = \mathbf{x}_{sv} \mathbf{x}_{sv}^T - 2\sum_{i=1}^{N} \alpha_i \mathbf{x}_i \mathbf{x}_{sv}^T + \sum_{i=1}^{N} \sum_{j=1}^{N} \alpha_i \alpha_j \mathbf{x}_i \mathbf{x}_j^T \tag{9}$$

where $\mathbf{x}_{sv}$ is the boundary support vector.

Kernel functions can be used in the SVDD, making it more flexible. A nonlinear mapping $\varphi$: $\mathbf{x} \rightarrow \mathbf{H}$ is introduced, where $\mathbf{H}$ represents a hypersphere. At this time, the optimization objective of the SVDD remains unchanged, as expressed in Equation (5). However, the constraint becomes

$$\|\varphi(\mathbf{x}_i) - \mathbf{a}\|^2 \le r^2 + \xi_i, \ i = 1, 2, \cdots, N \tag{10}$$

Similarly, Lagrangian factors $\alpha_i$ and $\beta_i$ are introduced to construct the Lagrangian function. Subsequently, the original problem becomes a quadratic programming problem, expressed as

$$\max_{\alpha} \sum_{i=1}^{N} \alpha_i K(\mathbf{x}_i, \mathbf{x}_i) - \sum_{i=1}^{N} \sum_{j=1}^{N} \alpha_i \alpha_j K(\mathbf{x}_i, \mathbf{x}_j)$$

$$\text{s.t.} \ 0 \le \alpha_i \le P, \sum_{i=1}^{N} \alpha_i = 1 \tag{11}$$

where $K(\cdot, \cdot)$ is the kernel function equivalent to the sample's inner product in the feature space. By solving Equation (12) to obtain the optimal solution of $\alpha$, the center of the obtained model hypersphere is expressed as follows:

$$\mathbf{a} = \sum_{i=1}^{N} \alpha_i \varphi(\mathbf{x}_i) \tag{12}$$

The radius $r$ is expressed as:

$$r^2 = K(\mathbf{x}_{sv}, \mathbf{x}_{sv}) - 2\sum_{i=1}^{N} \alpha_i K(\mathbf{x}_{sv}, \mathbf{x}_i) + \sum_{i=1}^{N} \sum_{j=1}^{N} \alpha_i \alpha_j K(\mathbf{x}_i, \mathbf{x}_j) \tag{13}$$

### 2.3. Multilayer Bi–LSTM Network

In deep learning, the input of each layer of a fully connected deep neural network and a convolutional neural network (CNN) is only related to the previous layer. In sequence samples, such as in natural language processing, speech recognition, text translation, and other tasks, the order of the samples is important. Therefore, to satisfy this requirement, a recurrent neural network (RNN) was derived. Research on RNN began in the 1980s, among which the bidirectional recurrent neural network (Bi–RNN) [40] and long–short–term memory (LSTM) are the most common. The Bi–LSTM used in this study combines the bi–directionality of the Bi–RNN with LSTM, retains the excellent learning ability of the LSTM for time series, and has the ability of the Bi–RNN to calculate backward. It is necessary to introduce the LSTM to better understand the Bi–LSTM.

### 2.3.1. LSTM

The LSTM network is a special RNN network that was proposed by researchers to satisfy performance requirements. By introducing gating units, the accumulated time–memory units are dynamically changed to improve the processing of time series. Figure 5 shows the basic structure of the LSTM unit [41].

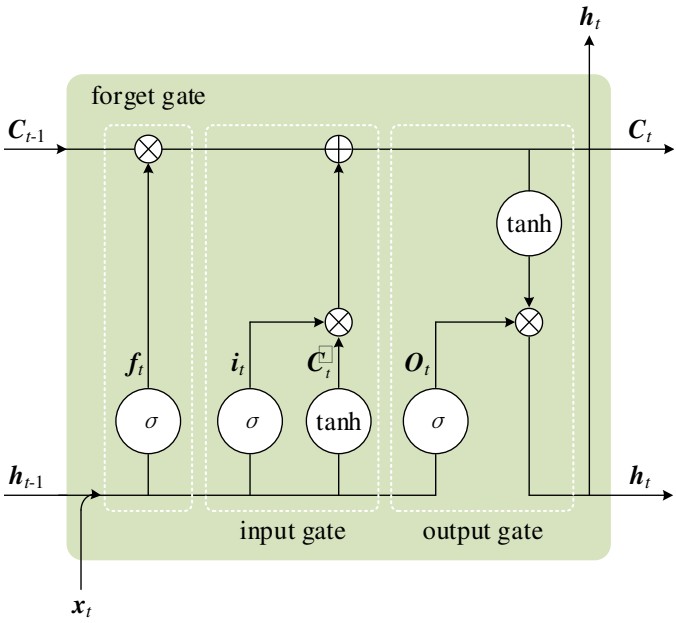

**Figure 5.** The basic structure of the LSTM unit.

In Figure 5, $x$ is the input layer, $h$ is the hidden network, and $C$ is the long–term memory unit. To realize the control and flow of storage memory, LSTM introduces three gate units—the forget, input, and output gates.

The first is a forget gate. It is used to determine the information to be retained in the state unit. It is expressed as follows:

$$f_t = \sigma\left(W_f[h_{t-1}, x_t] + b_f\right) \tag{14}$$

where $\sigma$ is a sigmoid function used to limit the retained elements to between 0 and 1; $W_f$ represents the weight–coefficient matrix of the forget gate; $[h_{t-1}, x_t]$ represents the concatenation of the previous time–step state and input at this moment; and $b_f$ represents the bias vector of the forget gate.

The input gate is used to determine the input information. It is expressed as follows:

$$\begin{cases} i_t = \sigma\left(W_i[h_{t-1}, x_t] + b_f\right) \\ \widetilde{C}_t = \tanh(W_c[h_{t-1}, x_t] + b_c) \end{cases} \tag{15}$$

The input gate is composed of the input layer element $x_t$ at the current moment and the hidden layer element $h_{t-1}$ at the previous moment through sigmoid transformation. The tanh function is used to scale the input information for the input gate to filter. The results from these two gating units are then combined and updated into the long–term memory unit, expressed as

$$C_t = f_t \circ C_{t-1} + i_t \circ \widetilde{C}_t \tag{16}$$

where $\circ$ is the Hadamard product [42]; $f_t \circ C_{t-1}$ is used to forget part of the information of the long–term memory state and prepare for the input of new information; and $i_t \circ \widetilde{C}_t$ is used to select the current input information and accumulate it in the long–term memory unit.

The final output result of the entire LSTM unit is controlled by the output gate. The long–term memory unit $C_t$ at the current moment is transformed using the tanh function. The final output is obtained by sequentially multiplying $O_t$ and its corresponding elements. It is expressed as follows:

$$\begin{cases} O_t = \sigma(W_o[h_{t-1}, x_t] + b_o) \\ h_t = O_t \circ \tanh(C_t) \end{cases} \tag{17}$$

Similar to the RNN, the LSTM uses the time–based backpropagation algorithm, which is backpropagation through time, to update the weight parameters. By introducing the gating unit, the LSTM network can process data with long–term dependencies more efficiently than an ordinary recurrent neural network.

### 2.3.2. Multilayer Bi–LSTM Network

LSTM can effectively mine the information from the front to the back of the time series; however, it fails to utilize the future information of the data. To address this problem, Bi–LSTM was proposed. It involves adding a layer of LSTM units, which is equivalent to recalculating the input sequence inversely. The final result is a combination of the results of the two–layer LSTM network. Figure 6 shows the Bi–LSTM structure.

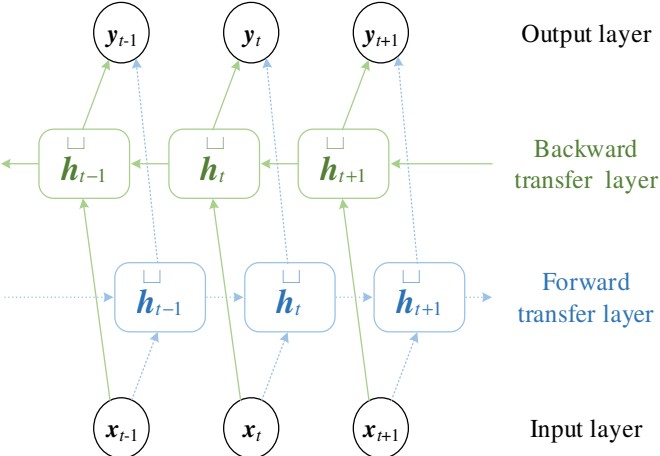

**Figure 6.** Schematic of Bi–LSTM network structure.

The output $H_t$ of the Bi–LSTM at time $t$ is expressed as

$$H_t = [\overrightarrow{h}_t, \overleftarrow{h}_t] \tag{18}$$

where $\overleftarrow{h}_t$ is the backward transfer layer, and $\overrightarrow{h}_t$ is the forward transfer layer. They are represented as

$$\begin{cases} \overrightarrow{h}_t = \overrightarrow{\mathrm{LSTM}}(\overrightarrow{h}_{t-1}, x_t, c_{t-1}), t \in [1, T] \\ \overleftarrow{h}_t = \overleftarrow{\mathrm{LSTM}}(\overleftarrow{h}_{t+1}, x_t, c_{t+1}), t \in [1, T] \end{cases} \tag{19}$$

where $x_t$ is the input at time $t$; $h$ and $c$ are the hidden layer state and memory unit state, respectively; and $T$ is the sequence length.

Multiple Bi–LSTMs are stacked and used to increase the learning ability of the model. In this new network structure, the input of the Bi–LSTM of each layer is the hidden layer output of the time step of the previous layer starting from the second layer. That is, the input of the multilayer Bi–LSTM unit at time $t$ in layer $p$ is $H_t^{p-1} = [\overleftarrow{h}_t^{p-1}, \overrightarrow{h}_t^{p-1}]$. Thus,

the forward propagation process of the multilayer Bi–LSTM at layer $p$ and time $t$ can be expressed as follows:

$$
\begin{cases}
f_t^p = \sigma\left(W_f^p[h_{t-1}^p, H_t^{p-1}] + b_f^p\right) \\
i_t^p = \sigma\left(W_i^p[h_{t-1}^p, H_t^{p-1}] + b_i^p\right) \\
o_t^p = \sigma\left(W_o^p[h_{t-1}^p, H_t^{p-1}] + b_o^p\right) \\
\widetilde{C}_t^p = \tanh\left(W_c^p[h_{t-1}^p, H_t^{p-1}] + b_c^p\right) \\
C_t^p = f_t^p \circ C_{t-1}^p + i_t^p \circ \widetilde{C}_t^p \\
h_t^p = o_t^p \circ \tanh\left(C_t^p\right)
\end{cases}
\tag{20}
$$

here, $f_t^p$, $i_t^p$, and $o_t^p$ represent the forget, input, and output gates of the multilayer Bi–LSTM unit in the $p$ layer at time $t$, respectively. $W_f^p$, $W_i^p$, $W_o^p$, and $W_c^p$ are the weight matrices of the forget, input, output gates, and memory unit of the Bi–LSTM unit in $p$–th layer, respectively. $b_f^p$, $b_i^p$, $b_o^p$, and $b_c^p$ are the corresponding bias vectors.

The last layer is marked as the $q$–th layer. The output of the multilayer Bi–LSTM at time $t$ can be expressed as

$$
y_t^q = \sigma\left(W_{h,y} h_t^q + b_{h,y}\right)
\tag{21}
$$

where $W_{h,y}$ is the weight matrix of the output $h_t^q$ of the Bi–LSTM unit in the $q$ layer at time $t$, and $b_{h,y}$ is the corresponding bias vector. Figure 7 shows the schematic diagram of the multilayer Bi–LSTM structure.

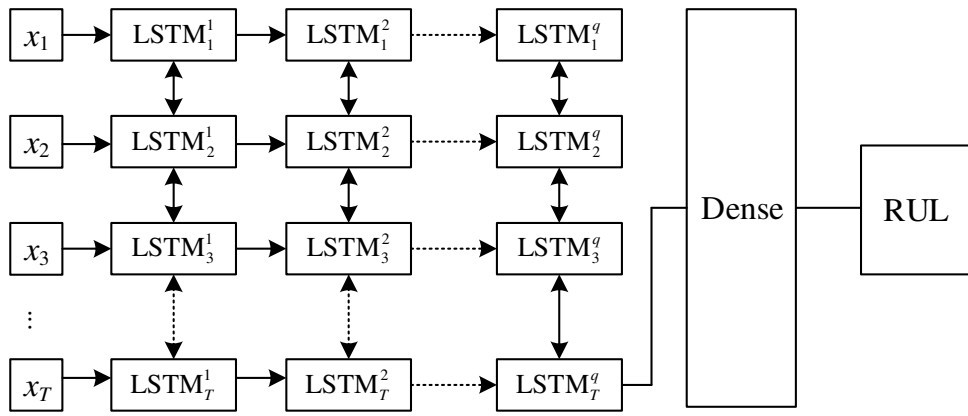

**Figure 7.** Schematic diagram of multilayer Bi–LSTM structure.

The predicted RUL can be obtained by connecting a dense layer at the last time step in the $q$–th layer of multilayer Bi–LSTM [43].

## 3. Experiment and Calculation

This section describes the test and the algorithm calculation flow. The flow chart is shown in Figure 8. Firstly, the collected data are extracted with features using DSAE, and after that, the features are fused by SVDD, and HI curves are plotted. Here, DSAE + SVDD is compared with the other two methods using the publicly available dataset. Finally, the gear pumps HI curves obtained by DSAE + SVDD are fed into a multilayer Bi–LSTM network for training the RUL prediction model, and the performance of the proposed method is verified using the test set data. The proposed method is compared with two other prediction methods.

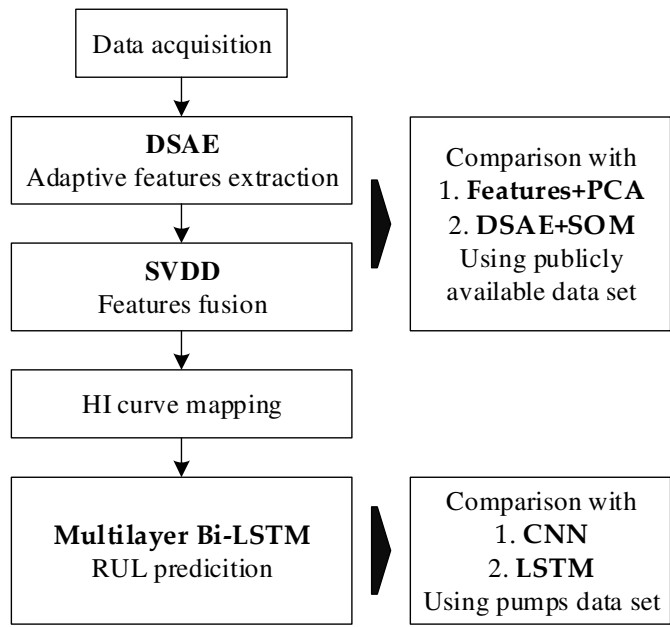

**Figure 8.** The flow chart of this section.

### 3.1. Full–Life Test of Gear Pump

The test used a gear pump full–life test bench to collect data. Figure 9 shows a schematic of the hydraulic system of the test bench. The test bench can simultaneously test four pumps. Each pump is equipped with vibration sensors in three directions—*x*, *y*, and *z*. In addition, monitoring sensors such as flow meters, pressure sensors, and temperature sensors are installed in the hydraulic system.

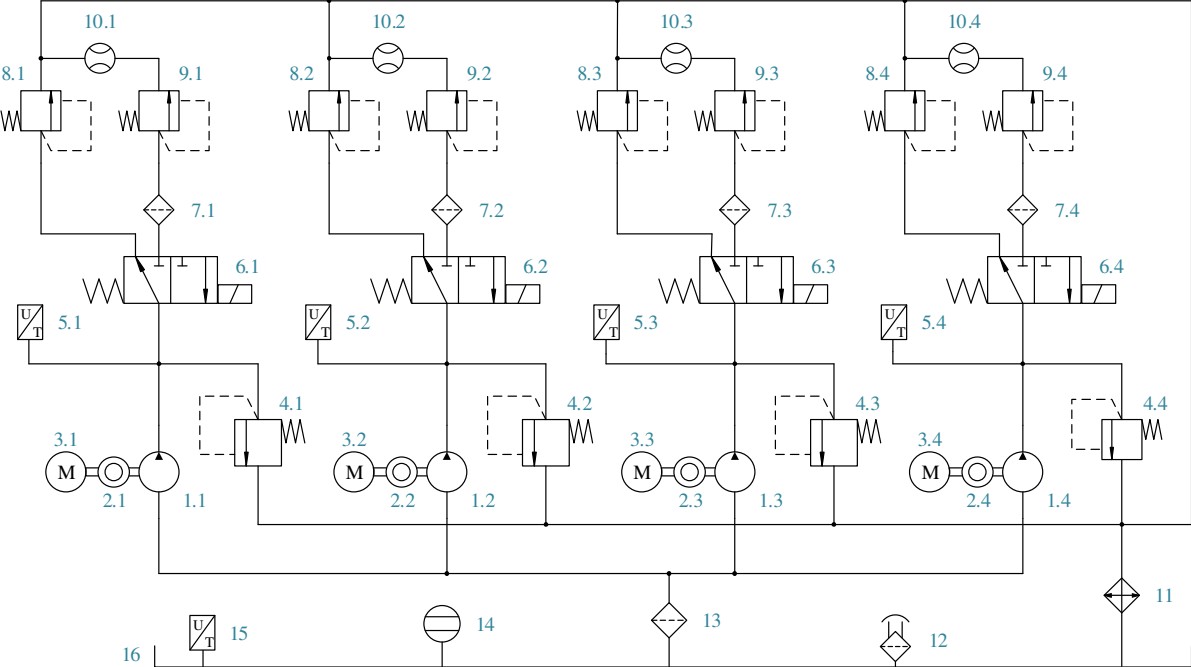

**Figure 9.** Schematic of the hydraulic system of the gear pump full–life test bench. 1, gear pump; 2, meter of rotating speed and torque; 3, electromotor; 4, relief valve; 5, pressure sensor; 6, solenoid directional valve; 7, high–pressure filter; 8, high–pressure relief valve; 9, low–pressure relief valve; 10, flow meter; 11, cooler; 12, air filter; 13, oil filter; 14, liquid level gauge; 15, temperature sensor; 16, tank.

During the test, the motor rotation speed was 1470 r/min. Four pumps were loaded simultaneously. Each pump had two circuits: a high–pressure circuit and a low–pressure (rated pressure) circuit. No flowmeters were installed in the high–pressure branch. When the gear pump operated on the high–pressure branch for 59 min and 40 s, solenoid directional valve 6 acted to adjust the working pressure to the rated value for 20 s and collect data in the last 2 s. The sampling frequency was set at 12 kHz. LabVIEW was used to control the cooler to ensure that the hydraulic oil temperature was between 40 °C and 50 °C to reduce the influence of the hydraulic oil temperature on the test. In addition, the test accelerated the degradation process of the gear pump by stepping up the load pressure level. The test bench was first operated at 23 MPa for 300 h. Subsequently, the pressure was increased to 25 MPa for 300 h. Finally, the pressure was increased to 27 MPa until the volumetric efficiency of the four gear pumps was lower than 70%. The test was completed when all hydraulic pumps failed. Table 1 lists the main components of the test bench and performance parameters.

**Table 1.** Test bench components and their performance parameters.

| Name | Model Number | Remarks |
|---|---|---|
| Gear pump | CBWF–304 | Rated pressure: 20 MPa, maximum pressure: 25 MPa, rated rotating speed: 2500 r/min, displacement: 4 mL/r. |
| Meter of rotating speed and torque | CYT–302 | Measurable torque range: 0–50 N·m, Measurable rotating speed range: 0–3000 r/min, accuracy: 0.5 FS. |
| Electromotor | Y90L–4B35 | Rated rotating speed: 1470 r/min, power: 3 kW. |
| Relief valve | DBDS6P1X/200 | Maximum pressure: 31.5 MPa, maximum flow: 80 L/min. |
| Pressure sensor | PU5400 | Maximum measurable value: 40 MPa. |
| Solenoid directional valve | 3WE6A50/G24 | Maximum flow: 60 L/min. |
| High–pressure filter | ZU–H40X30 | Maximum flow: 40 L/min, filter fineness: 30 μm. |
| Flow meter | MG015 | Maximum flow: 40 L/min. |
| Cooler | DEL–4 | Heat release: 0.3 kW/°C, flow range: 15–80 L/min. |
| Oil filter | TF–63X100F–Y | Nominal flow: 63 L/min, filter fineness: 100 μm, latus rectum: 25 mm. |
| Liquid level gauge | YWT–250 | Pressure range: 0.1–0.15 MPa. |
| DAQ card | NI PXIe–6363 | Analog acquisition channel: 32, resolution: 16–bit, maximum sampling rate: 2 MS/s |
| Temperature sensor | CWDZ11 | Measurable temperature range: −50–100 °C, accuracy: 0.5 FS |
| Acceleration sensor | YD–36D | Sensitivity: 0.002 V/m·s$^{-2}$, frequency response: 10 Hz–5 kHz (−3 dB), maximum acceleration: 2500 m/s$^2$, resolution: 0.01 m/s$^2$. |

This study developed a test and control program using LabVIEW. Figure 10 shows the front panel of the data acquisition program. It can perform real–time data monitoring of quantities such as pressure and temperature to ensure the normal operation of the system. Real–time status monitoring and data collection were performed simultaneously on the pressure, flow, torque, rotating speed, and vibration (in three directions) of the four tested pumps. LabVIEW is a graphical programming environment engineers use to develop automated research, validation, and production test systems. The program diagram of this data acquisition program is shown in Figure 11.

The failure mechanism of the tested gear pump is wear, which leads to an increase in internal leakage and a decrease in volumetric efficiency. The flow degradation curve of the tested gear pump is shown in Figure 12.

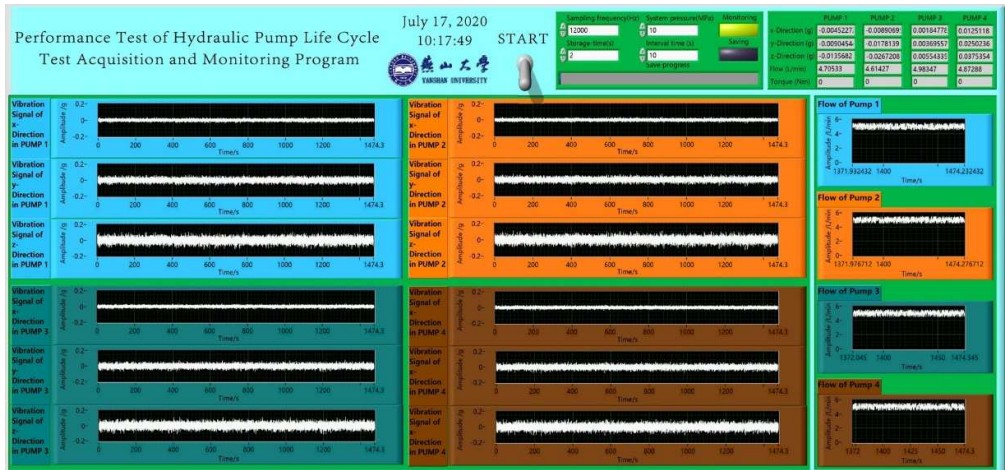

**Figure 10.** Data acquisition program front panel.

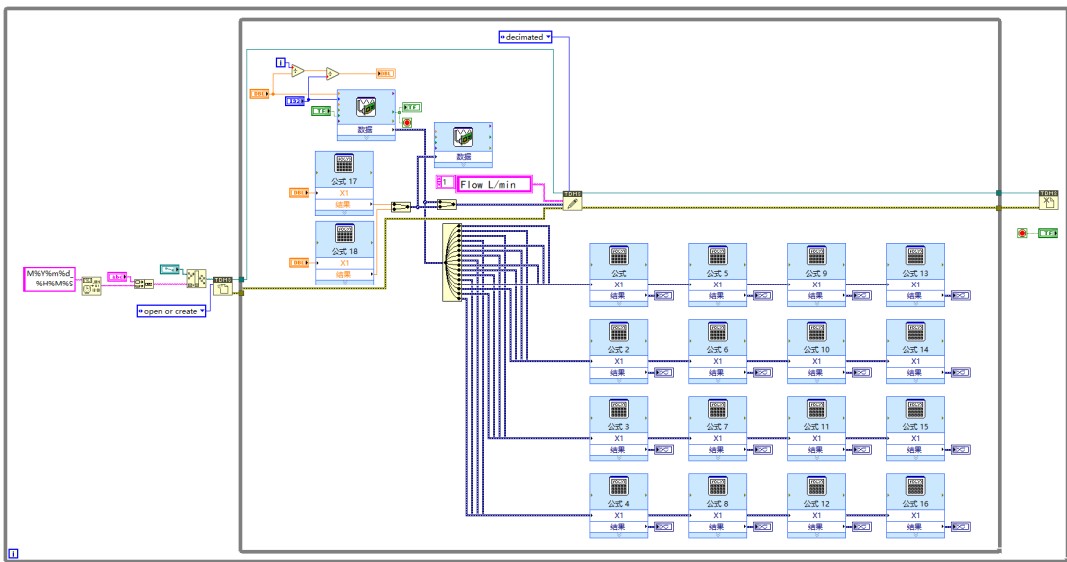

**Figure 11.** Data acquisition graphical program.

The output flow of the tested gear pump decreases significantly with the extension of running time. The flow rate of the gear pump decreases slowly in the early stage of operation and gradually accelerates in the later stage of operation until the failure threshold is reached.

According to [44], the life *L* of the gear pump is usually calculated with the bearing as the fault–sensitive part. The calculation formula is as follows:

$$
\begin{cases}
L = \frac{\lambda}{n}(\Delta p B D_e)^{-\frac{10}{3}} \\
\lambda = \frac{10^6}{60}\left(\frac{C}{0.15}\right)^{\frac{10}{3}}
\end{cases}
\tag{22}
$$

where *n* is the rotating speed of the gear pump; $\Delta p$ denotes the pressure difference between the inlet and outlet of the pump; *B* denotes the gear tooth width; $D_e$ denotes the diameter of the gear tip circle; and *C* is the basic dynamic load rating of the bearing.

According to Equation (22), the gear pump life obtained in this test was approximately 61.29–79.22% of the rated life.

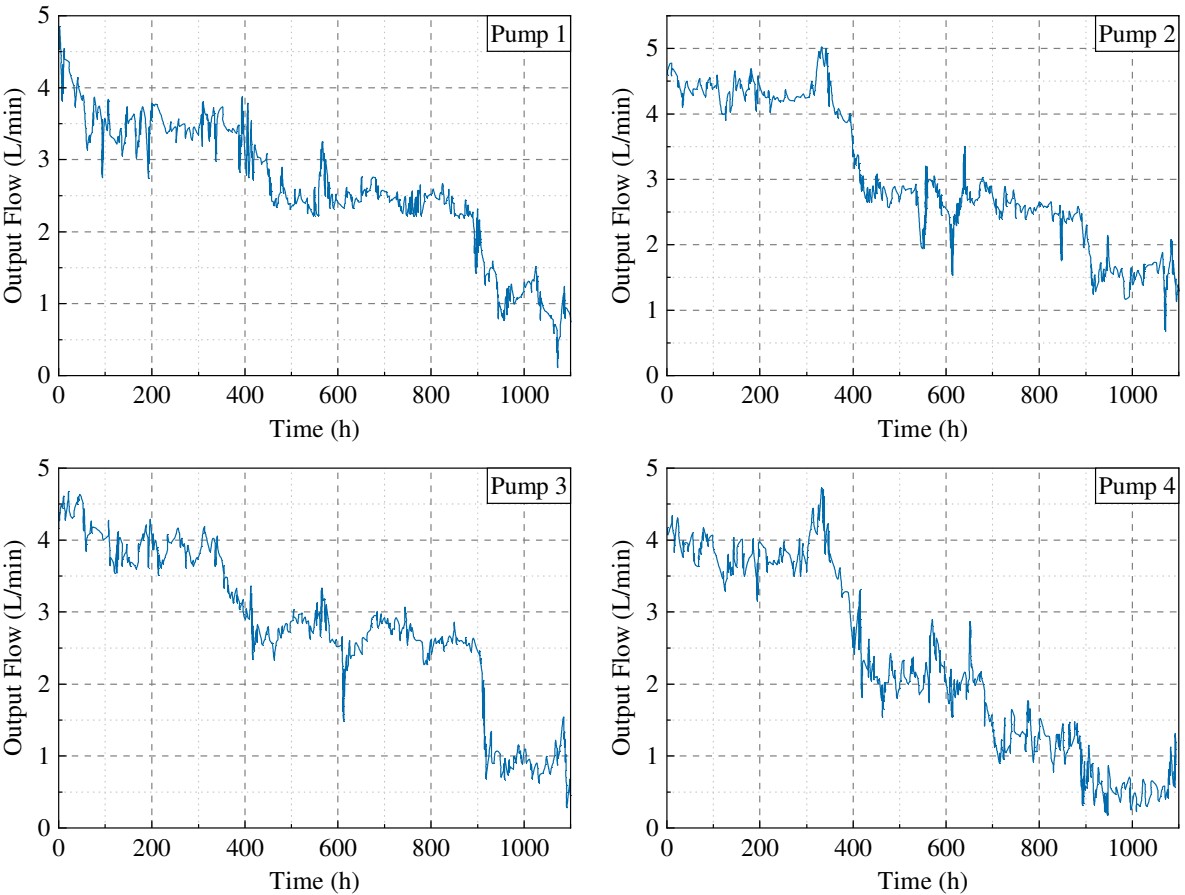

**Figure 12.** The flow degradation curve of the tested gear pumps.

### *3.2. Plotting the Health Indicator Curve by DSAE + SVDD*

As discussed in Section 2.1, the DSAE can extract more complex and abstract features by stacking multiple SAEs to extract input data layer by layer. Considering that the vibration signal contains rich information and is nonlinear and nonstationary, using only DSAE to extract a single feature cannot express its degradation characteristics well. Therefore, a feature fusion using SVDD was developed. The hypersphere was trained to obtain the center of the hypersphere through normal–state data in the early stage. Finally, the distance between the input data and the center of the hypersphere was calculated as the health indicator (HI) value. Figure 13 shows a flowchart of the HI curve drawing and algorithm evaluation.

Figure 13 shows that the vibration data sampled in the test were first subjected to the fast Fourier transform (FFT). Subsequently, the transformed data were normalized and divided into a training set and a test set. The training dataset was used to train the DSAE model. The hidden layer of the model was used as the feature set. The feature set was then inputted into the SVDD to train the hypersphere model. Subsequently, the test dataset was inputted into the trained DSAE + SVDD model. Finally, the HI value was obtained; the HI curve was constructed, and the algorithm was evaluated.

#### 3.2.1. Verifying the Proposed Method by the Public Dataset

To verify the effectiveness of the DSAE + SVDD method, the public bearing dataset from PHM2012 was selected to validate the method experimentally. In addition, this dataset was substituted into the other two methods, and the evaluation indexes of the calculated results were compared with the method used in this study. The other two methods used were as follows: (1) common characteristics of the time and frequency

domains with principal component analysis (PCA) and (2) DSAE with self–organizing mapping (SOM) [45].

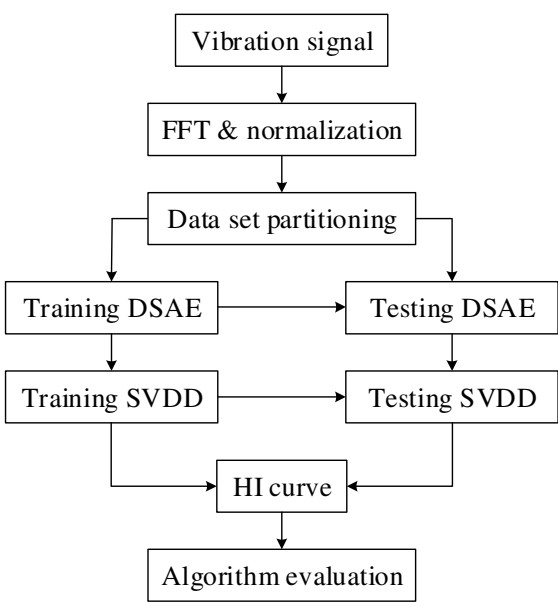

**Figure 13.** HI curve drawing and algorithm evaluation.

The PHM2012 dataset was provided by the FEMTO–ST Institute, and the experiment was performed on a PRONOSTIA test bench [46], as shown in Figure 14.

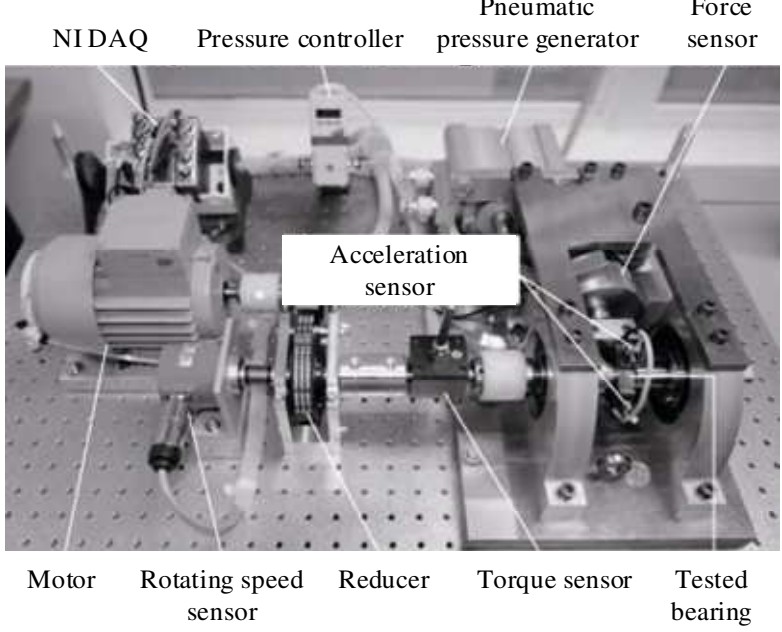

**Figure 14.** PRONOSTIA test bench.

The dataset included three different operating conditions. For each condition, two experimental bearing datasets, from beginning to failure, were provided to construct the degradation model, and 11 bearings were used to predict the RUL, as listed in Table 2. The vibration signal of this dataset was sampled at 25.6 kHz, with a sampling interval of 10 s. Each sample contained 2560 sampling points. The sampling time was 0.1 s.

**Table 2.** Description of the PHM2012 dataset.

| Category | Condition_1 | Condition_2 | Condition_3 |
|---|---|---|---|
| Training dataset | Bearing1_1<br>Bearing1_2 | Bearing2_1<br>Bearing2_2 | Bearing3_1<br>Bearing3_2 |
| Testing dataset | Bearing1_3<br>Bearing1_4<br>Bearing1_5<br>Bearing1_6<br>Bearing1_7 | Bearing2_3<br>Bearing2_4<br>Bearing2_5<br>Bearing2_6<br>Bearing2_7 | Bearing3_3 |

This study conducted experimental verification using the dataset of seven bearings from Condition_1, and the dataset was repartitioned. Table 3 lists the specific information for the repartitioned dataset. The data in the horizontal direction of the vibration signal were used for the operation. The bearing failure process could not be accurately controlled in the experiment; therefore, when the vibration signal was above 20 g, it was considered a bearing failure. This rule left approximately 10% of the data at the end of the service life of unused bearings.

**Table 3.** Dataset division and usage.

| Category | Dataset | Number of Used Samples | Number of All Samples |
|---|---|---|---|
| Training dataset | Bearing1_1<br>Bearing1_2<br>Bearing1_3<br>Bearing1_4<br>Bearing1_5 | 2660<br>826<br>2329<br>1055<br>2217 | 2803<br>871<br>2375<br>1428<br>2463 |
| Testing dataset | Bearing1_6<br>Bearing1_7 | 2080<br>2033 | 2448<br>2259 |

The degradation modeling process can be described as follows.

Step 1: Preprocessing the vibration data. First, FFT was performed on the original data, and the amplitude of the vibration data in the spectrum was normalized. The normalization method is expressed as follows:

$$X^* = \frac{X - X_{\min}}{X_{\max} - X_{\min}} \tag{23}$$

where $X_{\min}$ and $X_{\max}$ are the minimum and maximum values of the training set after FFT, respectively. The testing set was processed using the same normalization method. Figure 15 shows a three−dimensional spectrogram of the vibration data after normalization.

Figure 15 shows that after 400 min, there was an increase in several frequency bands when the bearing was used. The most apparent increase was observed near the narrow band at 1250 Hz.

Step 2: DSAE training. The training data were inputted into the SAE for training. Each SAE was trained 100 times. At completion, the encoded part was retained, and the output was used as the input for the next SAE. Finally, the entire stacked network was globally finetuned 20 times using the BP algorithm. This improved the generalization ability of the model and yielded a better feature representation.

The first bearing vibration data were used as an example for illustration. The length of each sample was 2048 after the FFT; that is, the number of neurons in the input and output layers was 2048. Therefore, the network structure of the model was 2048–1000–500–100–500–1000–2048. The coding part comprised three stacked SAEs; the structures were 2048–1000–2048, 1000–500–1000, and 500–100–500 to obtain 100–dimensional deep features. Owing to space limitations, the first 10 dimensions were selected from the results, and the curves

were obtained after processing by adjacent averaging with a step length of 50, as shown in Figure 16.

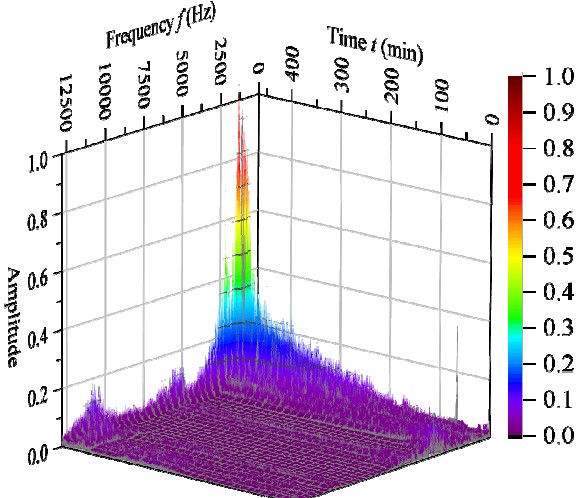

**Figure 15.** 3D diagram of the first bearing vibration data in time–frequency domain.

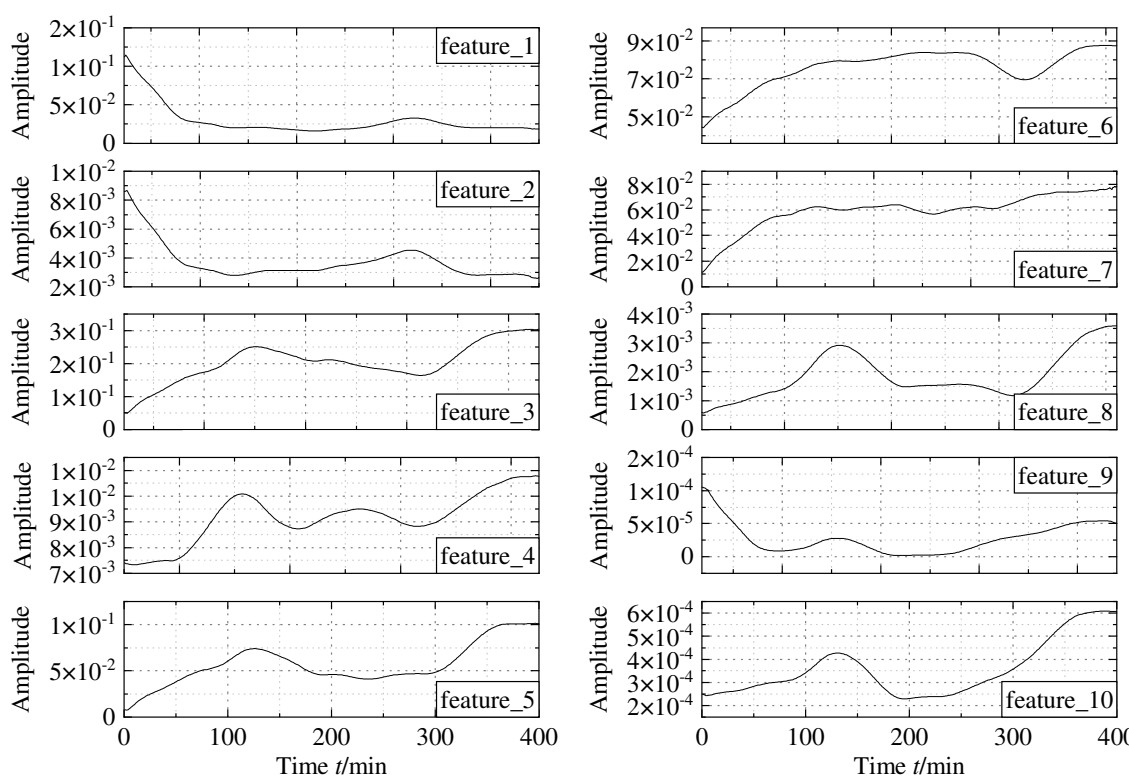

**Figure 16.** Curves of the first 10 characteristics after smoothing.

Figure 16 shows that the features extracted by the DSAE have good monotonicity overall, effectively representing the degradation process of the bearing performance.

Step 3: SVDD model training. Approximately 15% of the data of each bearing prophase in the training set were selected and inputted to the DSAE to extract high–dimensional features. They were used as the input to the SVDD model. It can be assumed that these early bearing data reflect the normal state; thus, the hypersphere center was obtained. The features extracted from the full–life data of each bearing were then inputted into the SVDD model. The kernel function of the SVDD was chosen as the Gaussian kernel function, the penalty factor $P$ was 1, and the Gaussian width $\sigma$ was 0.003. The distance between

each sample and the hypersphere center was obtained and used as the bearing HI at the corresponding time point. The first five bearing datasets used as the training set were inputted to the DSAE + SVDD model to obtain the HI curves of each bearing, as shown in Figure 17.

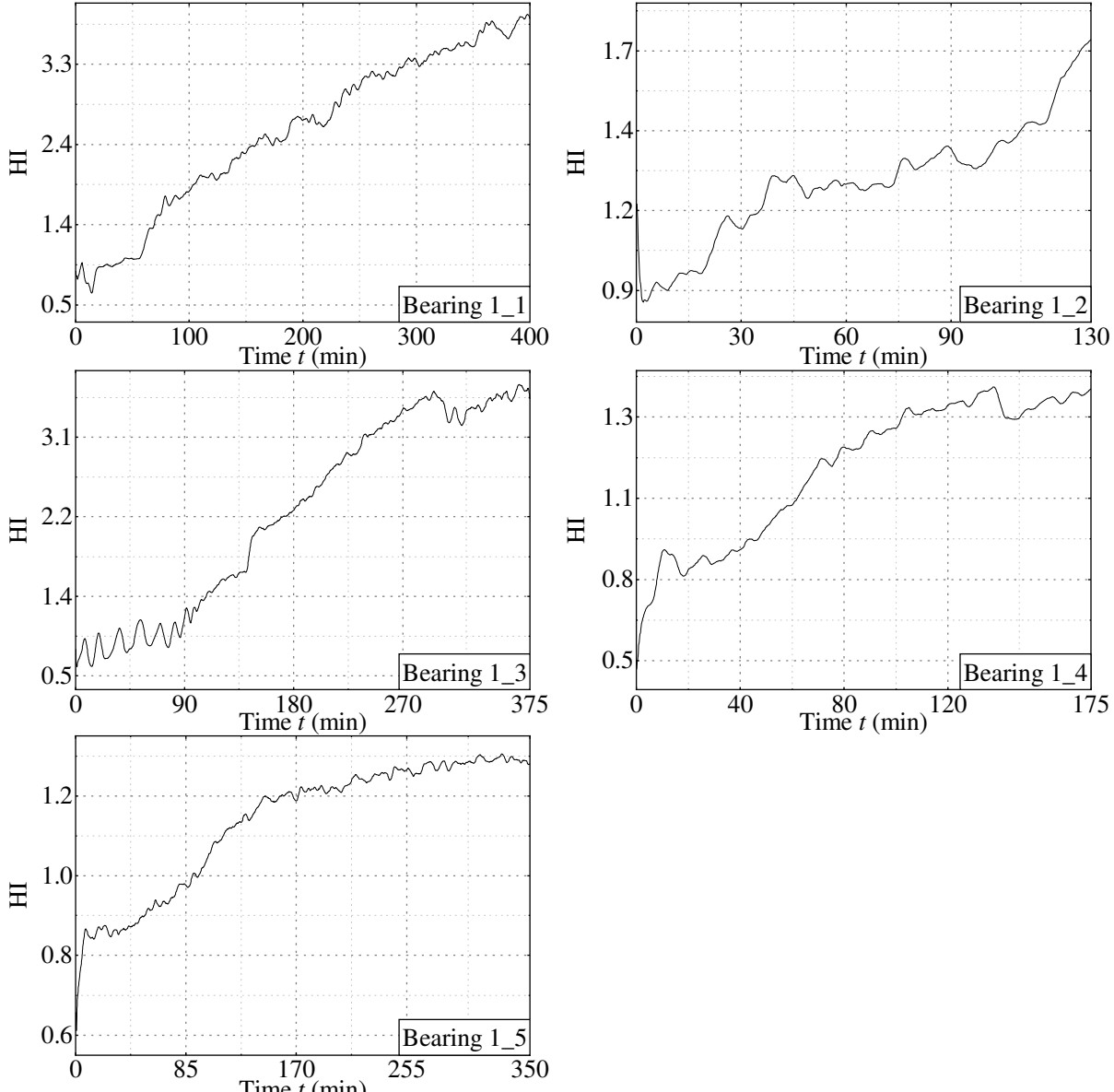

**Figure 17.** HI curves of the training bearings.

Figure 17 shows that the HI curve gradually increases with time, exhibiting monotonicity and reflecting the degradation process of the bearings.

Step 4: Validation using testing datasets. The data from the testing datasets were sequentially inputted into the trained DSAE and SVDD models. Subsequently, the HI curves for each bearing in the testing datasets were obtained, as shown in Figure 18.

Figure 18 shows that the HI curves obtained from the two testing datasets exhibited a clear upward trend after the trained DSAE + SVDD model operation. This is consistent with the expected results and proves that the method can effectively reflect the degradation of equipment.

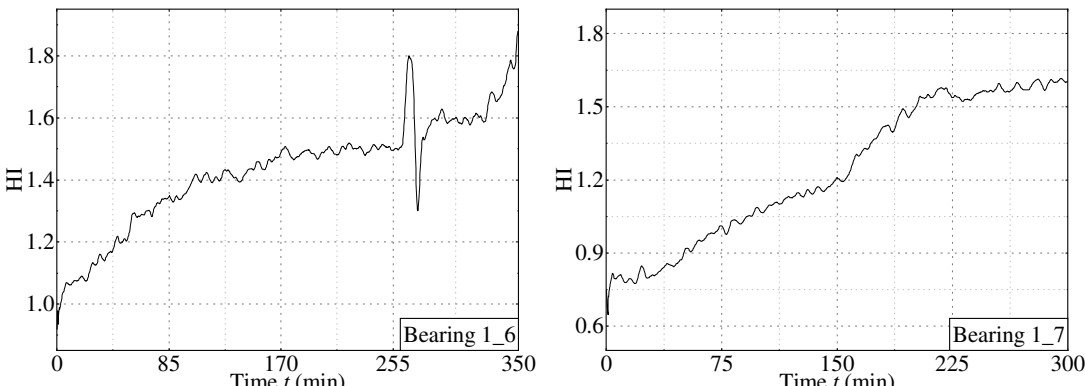

**Figure 18.** HI curves of the testing bearings.

Step 5: Algorithm evaluation. Two methods were selected for comparison to analyze the proposed method.

Method 1: Feature extraction + PCA. The features used included the following: (1) time–domain features (root–mean–square, peak–to–peak value, variance, kurtosis, skewness, waveform indicator, crest indicator, and margin indicator); (2) frequency domain features (root mean square frequency, gravity frequency, frequency standard deviation, and frequency variance); and (3) eight sub–band energies obtained by decomposing a three–layer wavelet packet [47]. The 21 features obtained were reduced in dimensionality by the PCA to obtain the HI curve of the bearing.

Method 2: DSAE + SOM. Similar to the proposed method, the features of the normal state data were extracted by the DSAE and inputted to the SOM. The number of nodes in the input layer of the SOM was 100, which was the dimension of the features extracted by the DSAE. The output layer was a two–dimensional planar array. In the topology layer, the distance between each node and the input data was calculated. The smallest distance was selected as the HI to draw the curve.

This study selected the monotonicity and trend indices [48,49] to quantitatively analyze the performance advantages and disadvantages of the three methods. The monotonicity index, $V_{mon}$, was used to measure the monotonic trend of the HI curve. An ideal HI curve should exhibit a monotonically increasing or decreasing trend because the degradation process of mechanical equipment, such as bearings and their components, is irreversible. The trendability index, $V_{tre}$, reflects the correlation between the degradation trend of the HI curve and running time. The closer the indicator is to 1, the better the HI curve trend. The two expressions are defined as follows:

$$V_{mon} = \left| \frac{Num\ of\ \Delta F > 0}{T - 1} - \frac{Num\ of\ \Delta F < 0}{T - 1} \right| \tag{24}$$

$$V_{tre} = \frac{\left| \sum\limits_{t=1}^{T} (H_t - \hat{H})(l_t - \hat{l}) \right|}{\sqrt{\sum\limits_{t=1}^{T} (H_t - \hat{H})^2 \sum\limits_{t-1}^{T} (l_t - \hat{l})^2}} \tag{25}$$

where $\Delta F$ is the difference between two adjacent points in the HI curve, and $T$ is the number of full–life data points of the bearing. $H_t$ is the HI value corresponding to the $t$–th data point, $l_t$ is the time number at which the $t$–th data point is located, $\hat{H}$ is the average of the HI curve value, and $\hat{l}$ is the data point serial number.

Three methods were used to construct the HI curves for the seven bearing datasets under Condition_1. The constructed curves were evaluated using the evaluation indexes. Table 4 lists the evaluation results.

**Table 4.** Evaluation results of HI curves constructed by different methods.

| Dataset | Features + PCA | | DSAE + SOM | | DSAE + SVDD | |
|---|---|---|---|---|---|---|
| | $V_{mon}$ | $V_{tre}$ | $V_{mon}$ | $V_{tre}$ | $V_{mon}$ | $V_{tre}$ |
| Bearing1_1 | 0.13 | 0.86 | 0.17 | 0.66 | **0.21** | **0.96** |
| Bearing1_2 | 0.07 | 0.19 | 0.03 | 0.31 | **0.27** | **0.94** |
| Bearing1_3 | 0.13 | 0.73 | 0.14 | 0.64 | **0.34** | **0.97** |
| Bearing1_4 | 0.09 | 0.91 | 0.16 | **0.95** | **0.41** | 0.94 |
| Bearing1_5 | 0.16 | 0.91 | 0.17 | **0.92** | **0.19** | 0.92 |
| Bearing1_6 | 0.04 | 0.19 | 0.11 | 0.82 | **0.20** | **0.93** |
| Bearing1_7 | 0.06 | 0.56 | 0.08 | 0.29 | **0.21** | **0.96** |

The optimum values are highlighted in boldface.

Table 4 shows that the monotonicity indices of the seven HI curves constructed by the DSAE + SVDD method were better than those constructed by the other two methods. The trendability indices of six of the seven curves were also the best in the same group, proving that the method is superior.

3.2.2. Processing Gear Pump Data Using the Proposed Method

After evaluating the DSAE + SVDD method using a public dataset, the gear pumps vibration signals are described in Section 3.1 and were processed using this method. The specific steps are as follows:

Step 1: Data preprocessing. The data were divided into training and testing sets. The gear pump vibration signal amplitude effectively reflects its degradation trend; therefore, all the life data were normalized in the frequency domain. The process is expressed in Equation (23). In addition, time–domain normalization was performed on the rotating speed, torque, and pressure signals. This reduced the impact of excessive values on model training and increased the model convergence speed.

Step 2: Training the DSAE model. First, a single SAE was trained using unsupervised greedy layer–by–layer training with a training count of 50 and optimized using the Adam algorithm with an initial learning rate of 0.001. Finally, multiple SAEs were stacked and globally finetuned using the BP algorithm 10 times. In the gear pump life degradation test, the length of the data obtained from each sampling was 24,000. The first 1200 data points were intercepted as a sample to improve the operation efficiency. The DSAE structure used was 1200–600–150–30–150–600–1200, consisting of three SAEs stacked to extract 30–dimensional features.

Step 3: SVDD model training. The 30 features obtained from the normal state samples and normalized average values of the rotating speed, torque, and pressure form a 33–dimensional dataset of the normal state. This was substituted to the SVDD model for training to obtain the hypersphere center. The kernel function was chosen as Gaussian with a penalty factor $C$ of 1 and a Gaussian width $\sigma$ of 0.03.

Step 4: Validation using testing datasets. After model training, all data were inputted into the DSAE and SVDD models. Subsequently, the HI curves of the gear pumps were obtained by calculating the distance between each sample and the hypersphere center. The obtained curves were smoothed. Figure 19 shows the results.

Figure 19 shows that the HI curves of the four testing pumps exhibited an evident upward trend. These results demonstrate that the DSAE + SVDD method accurately models the performance degradation of gear pumps.

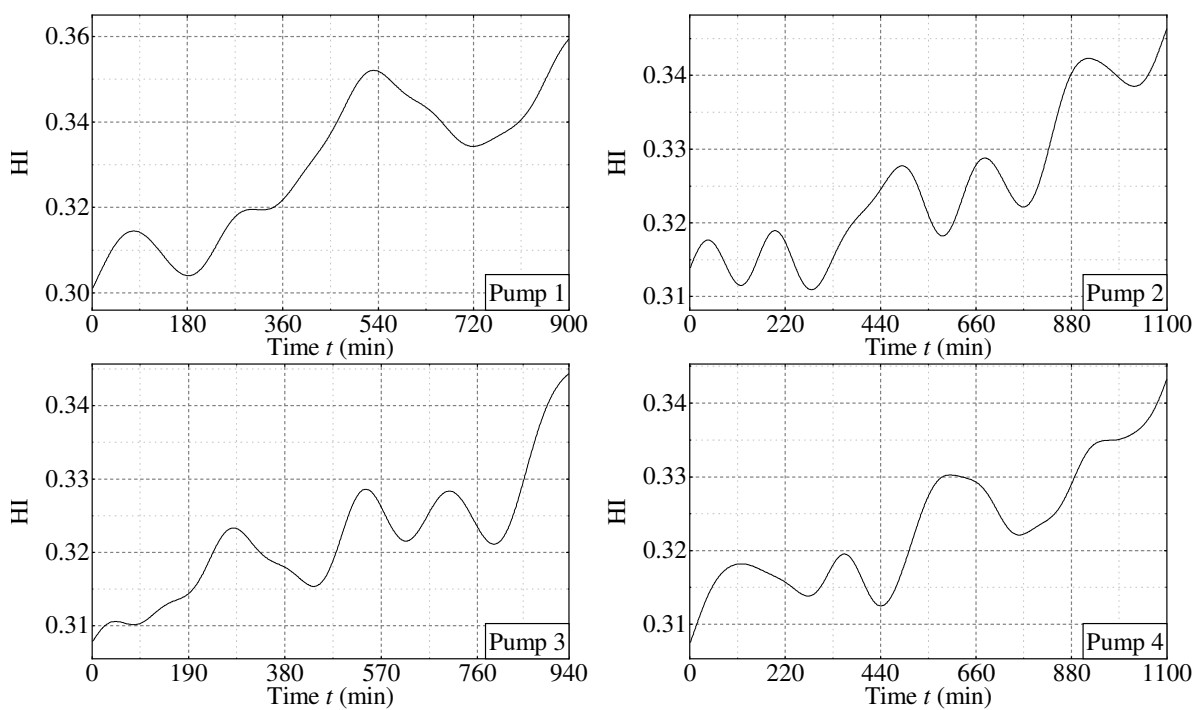

**Figure 19.** HI curves of gear pumps.

*3.3. Predicting RUL Using Multilayer Bi–LSTM*

In the RUL prediction process, the gear pump HI values obtained in Section 3.2.2 were normalized and divided into multiple samples. Thereafter, samples of the corresponding structures were constructed and labeled according to the requirements of the multilayer Bi–LSTM network. Finally, the processed samples were substituted into the multilayer Bi–LSTM network to complete the training and testing tasks. The specific process is as follows:

Step 1: Partitioning and normalization of the datasets. First, the HI values of the gear pumps were normalized by the z–score to reduce the effect of an excessive data discrepancy. The transformation function is as follows:

$$x^* = \frac{x - \mu}{\sigma} \tag{26}$$

where $\mu$ and $\sigma$ are the mean and standard deviation of the sample data, respectively. The datasets were then divided, and the datasets of pumps 1 and 2 were selected as the training sets. Data from the remaining two pumps were used as the test sets.

Step 2: Constructing and labeling the samples. The input format of the multilayer Bi–LSTM network is [BatchSize, TimeSteps, FeatureDims], where BatchSize is the number of batch samples, TimeSteps is the time step, and FeatureDims is the feature dimension. However, the HI curve is a one–dimensional sequence; therefore, the time step needs to be specified, and each HI sequence is segmented to construct a sample set. Assuming that the HI curve is drawn from $l$ HI values, the curves can be used to construct $l$–TimeSteps + 1 samples. The obtained samples are TimeSteps–dimensional column vectors. The label for each sample is the RUL value corresponding to the last HI value in the time step. Finally, all sample label values were normalized to increase the convergence speed of the model training.

Step 3: Multilayer Bi–LSTM model training. The training datasets were inputted into the model for training. The number of model layers and neurons in each layer were determined using the grid search algorithm. To study the influence of the number of layers on the prediction results, the Bi–LSTM model with one to five layers was selected for the test, the HI of pump 4 was selected as the test data, and the root mean square error (RMSE)

of the test results was used as an indicator. Figure 20a shows the RMSE as a function of the number of layers. The same dataset was used as an example to explore the time step effect. Two–layer Bi–LSTM models with time steps of 5, 10, 15, 20, 25, 30, and 35 were tested and analyzed. Figure 20b shows the RMSE with the step size.

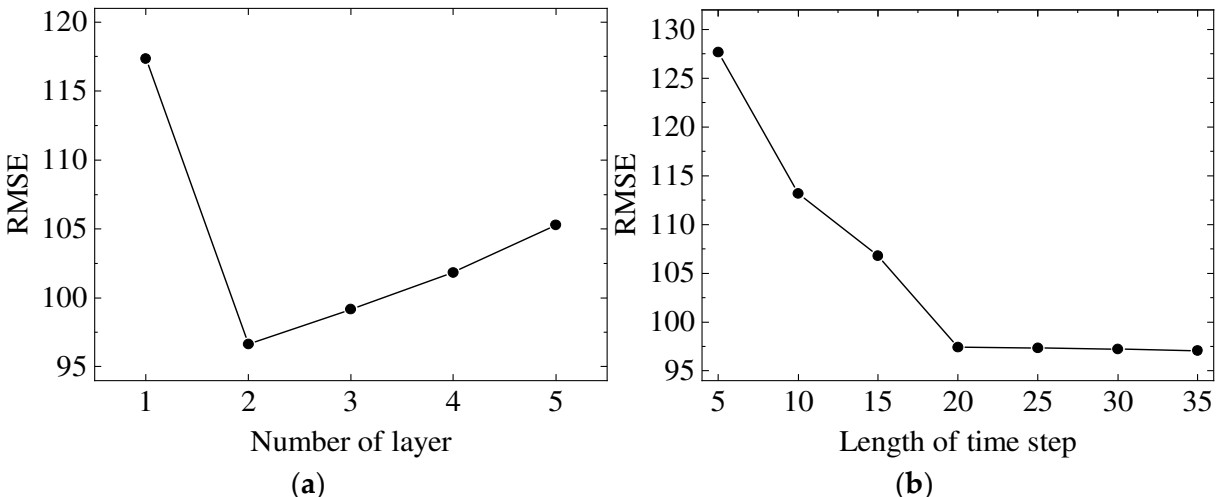

**Figure 20.** Test result error curves. (**a**) Description of RMSE of test results under different layers; (**b**) description of RMSE of test results under different time steps.

Figure 20a shows that the single–layer Bi–LSTM model had the largest RMSE value of 117.01, indicating that it had the worst prediction accuracy in the test range. The two–layer Bi–LSTM model had the smallest RMSE value of 96.43, indicating that this structure provided the best prediction. As the number of Bi–LSTM layers increased, the RMSE value of the prediction results also increased. In addition, the training time increased with an increase in the number of stacked Bi–LSTM layers. Therefore, the number of Bi–LSTM layers in the model was set to two.

Figure 20b shows that the RMSE values of the test results decreased rapidly as the time step increased. When the time step was 20, the RMSE values decreased slowly and almost converged to a constant value. Similarly, the training time increased as the time window increased. Therefore, the time step was set to 20 in subsequent experiments.

Finally, the number of Bi–LSTM layers in the model was determined to be 2, the time step was 20, the maximum remaining lifetime was set to 80% of the actual lifetime [50], and the constructed model was trained using the HI data of pump 1 and pump 2.

Step 4: Validation using testing datasets. The testing datasets obtained from step 2 were inputted into the trained multilayer Bi–LSTM network to obtain the prediction results and final RUL prediction values. Figure 21 shows the RUL prediction curves for pumps 3 and 4.

In practical applications, the predicted results of equipment in late life are often valued. Therefore, in this study, the comparison between the predicted and real values was performed by focusing on the last 20% of the lifetime period, and the first 80% was neglected. In addition, this study used CNN [51,52] and LSTM methods with the same datasets and calculated the evaluation indices to compare the performance with the methods described in this paper. The evaluation indices selected in this study included the mean absolute error (MAE), RMSE, and normalized mean squared error (NMSE).

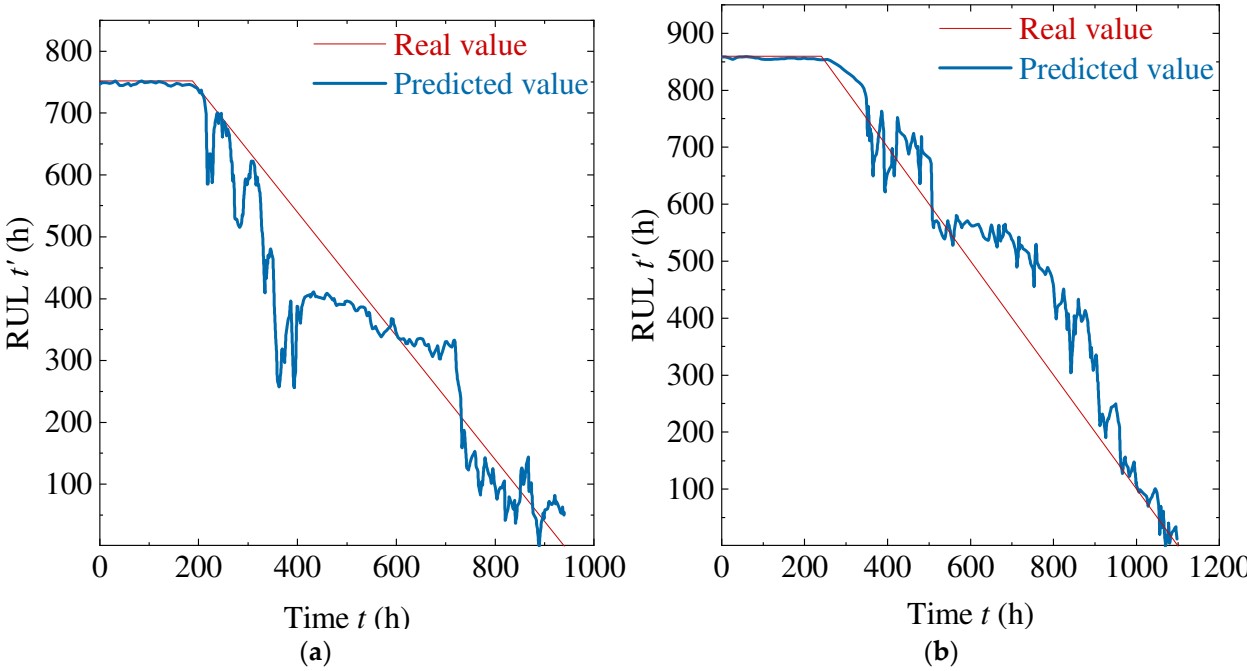

**Figure 21.** Predicted and actual RUL curves. (**a**) Description of curves for pump 3; (**b**) description of curves for pump 4.

The MAE is the average of the absolute value of the error between the predicted and true values for all moments. It can accurately reflect the magnitude of the prediction error because it has a cumulative operation. The MAE is given by

$$MAE = \frac{1}{N}\sum_{i=1}^{N}|\hat{x}_i - x_i| \tag{27}$$

The RMSE is also known as the standard error. Because it is more sensitive to the point where the prediction difference is larger, it can effectively reflect the accuracy of the prediction and is expressed as

$$RMSE = \sqrt{\frac{1}{N}\sum_{i=1}^{N}(\hat{x}_i - x_i)^2} \tag{28}$$

The NMSE is the ratio of the mean square error to the variance, reflecting the relationship between the prediction deviation and oscillation strength of the true value, expressed as follows:

$$\begin{cases} NMSE = \frac{1}{\sigma^2 N}\sum_{i=1}^{N}(\hat{x}_i - x_i)^2 \\ \sigma^2 = \frac{1}{N}\sum_{i=1}^{N}(x_i - \bar{x})^2 \end{cases} \tag{29}$$

Table 5 lists the computed prediction performances of the proposed multilayer Bi–LSTM algorithm and comparison algorithms.

Table 5 shows that the prediction performance of the proposed method on pump 3 was significantly better than that of the other two algorithms in terms of the three evaluation indices. For pump 4, the RMSE of the prediction results obtained by the CNN was smaller than that of the proposed method. The other two evaluation indices showed that the proposed method was better. Therefore, the method proposed in this study can accurately predict the RUL of gear pumps.

**Table 5.** Evaluation indexes of the multilayer Bi–LSTM and the comparison algorithms.

| Algorithm | Evaluation Index | Pump 3 | Pump 4 |
|---|---|---|---|
| CNN | MAE | 119.65 | 55.35 |
| | RMSE | 138.70 | **69.30** |
| | NMSE | 6.10 | 1.43 |
| LSTM | MAE | 110.00 | 63.82 |
| | RMSE | 138.45 | 79.38 |
| | NMSE | 6.08 | 3.12 |
| Multilayer Bi–LSTM | MAE | **38.78** | **52.82** |
| | RMSE | **43.17** | 73.37 |
| | NMSE | **0.76** | **1.07** |

The optimum values are highlighted in boldface.

## 4. Conclusions

This study developed a RUL prediction method for gear pumps. First, the multi–dimensional features of the vibration signal were extracted from the DSAE with a structure of 1200–600–150–30–150–600–1200 for the accelerated life test data of the gear pump. Subsequently, these features and the average of pressure, rotating speed, and gear pump torque at the corresponding time were fused by the SVDD to obtain the HI value at each moment. Finally, the double–layer Bi–LSTM (time step of 20) was used to complete the training model of the gear pump life prediction, and the test datasets were used for prediction testing. The following conclusions were drawn:

(1) The DSAE + SVDD method used in this study effectively constructed the HI curve of equipment, including the bearing and gear pump. This method is versatile. The entire development process was performed in an unsupervised condition, overcoming the subjectivity associated with artificial feature selection;

(2) Relying on the PHM2012 public bearing datasets, the DSAE + SVDD was compared with feature extraction + PCA and DSAE + SOM. The results show that the HI curves constructed by DSAE + SVDD had better trends and monotonicity indices than those of the two comparison algorithms;

(3) An RUL prediction method for multilayer Bi–LSTM for gear pumps was proposed. The results indicated that the method effectively predicted the RUL of a hydraulic gear pump in the later stages of life;

(4) Based on the gear pump–accelerated life degradation dataset, the multilayer Bi–LSTM method was compared with the CNN and LSTM methods. The results indicated that the three evaluation indexes of MAE, RMSE, and NMSE obtained by the multilayer Bi–LSTM were better than those of the comparison algorithms, proving the superiority of the multilayer Bi–LSTM in terms of life prediction ability.

The proposed method can accurately predict the RUL of gear pumps; however, there are still shortcomings. First, the proposed method in this paper is based on offline data for prediction, and there are already researchers who use real–time data for the prediction of the RUL of bearings. This real–time approach can be borrowed and applied to hydraulic pumps. In addition, the vibration sensors must be attached to the object under test to collect the signal, which can be troublesome in some cases. The noise generated by the vibration can be collected using a non–contact method, which will expand the application of fault diagnosis and RUL prediction techniques. In addition, there are other advantages to using noise as an analysis signal but not described here. Finally, in this paper, HI value calculation and RUL prediction are treated as two separate units. Thereafter, an attempt can be made to assemble these two parts into a unified model with the original signal as the input to obtain the prediction results directly.

**Author Contributions:** Conceptualization, P.Z. and X.S.; methodology, X.S.; software, X.S.; validation, P.Z., W.J. and X.S.; formal analysis, P.Z.; investigation, P.Z.; resources, W.J.; data curation, X.S.; writing—original draft preparation, P.Z.; writing—review and editing, W.J.; visualization, P.Z.;

supervision, W.J.; project administration, W.J.; funding acquisition, W.J. and S.Z. All authors have read and agreed to the published version of the manuscript.

**Funding:** This research has been supported by the National Natural Science Foundation of China, China (Grant No. 52275067 and Grant No. 51875498) and Province Natural Science Foundation of Hebei, China (Grant No. F2020203058 and Grant No. E2018203339).

**Institutional Review Board Statement:** Not applicable.

**Informed Consent Statement:** Not applicable.

**Data Availability Statement:** The data presented in this study are available on request from the corresponding author.

**Conflicts of Interest:** The authors declare no conflict of interest.

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
