# Peer review of "Remaining Useful Life Prediction of Gear Pump Based on Deep Sparse Autoencoders and Multilayer Bidirectional Long–Short–Term Memory Network"

_processes, doi:10.3390/pr10122500_

Round 1
Reviewer 1 Report
This paper is a continuation of the article that should be cited: Wang, C.; Jiang, W.; Yue, Y.; Zhang, S. Research on Prediction Method of Gear Pump Remaining Useful Life Based on DCAE and Bi-LSTM. Symmetry 2022, 14, 1111. https://doi.org/10.3390/sym14061111. However, it is a new approach through the prism of neural networks.
However, the reference time of 400 minutes and 1100 seems too short to me.
Author Response
Dear Reviewers:
Thank you for your letter and for the reviewers’ comments concerning our manuscript entitled “Remaining useful life prediction of gear pump based on deep sparse autoencoders and multilayer bidirectional long short-term memory network” (ID: processes-1984081). Those comments are all valuable and very helpful for revising and improving our paper, as well as the important guiding significance to our research. We have studied the comments carefully and have made corrections which we hope meet with approval. Revised portions are marked by the “Track Changes” function in the paper. The main corrections in the paper and the responses to the reviewer’s comments are as flowing:
Comment_1: This paper is a continuation of the article that should be cited: Wang, C.; Jiang, W.; Yue, Y.; Zhang, S. Research on Prediction Method of Gear Pump Remaining Useful Life Based on DCAE and Bi-LSTM. Symmetry 2022, 14, 1111. https://doi.org/10.3390/sym14061111. However, it is a new approach through the prism of neural networks.
Response: We are very sorry for our negligence. We communicate closely with several authors of this literature but missed this article in the writing process, which we have added to our references after careful reading.
Comment_2: The reference time of 400 minutes and 1100 seems too short to me.
Response: In our article, the gear pump life was set at 1100 hours, a significantly lower value than the theoretical gear pump life. This occurred because the test useed a gradual increase in load to accelerate the gear pump's degradation rate. We added the gear pump's output flow curve to section 3.1 of the article as a reflection of the degradation. Similarly, a similar approach was used to shorten the life of the bearing in the published data set.
Special thanks to you for your good comments.
We tried our best to improve the manuscript and made some changes in the manuscript. These changes will not influence the content and framework of the paper. And here we did not list the changes but marked them by the “Track Changes” function in the revised paper.
We appreciate for your warm work and hope that the correction will meet with approval.
Once again, thank you very much for your comments and suggestions.
Sincerely yours,
Pei-yao Zhang, PhD

Reviewer 2 Report
Recommendations for the authors:
a) to avoid the modeling and simulation of the complex systems without consulting a high-level design engineer;
b) to avoid the "chains" of algorithms without individual validation using wide accepted models;
c) to study the new types of sensors without contact which can give precious information concerning each component of the studied assembly; for example, thirty years ago the directional noise probes developed by LMS Company from Belgium were intensively developed for identifying the dominant noise generation components and the wear degree of different rotary machines components.
Author Response
Dear Reviewers:
Thank you for your letter and for the reviewers’ comments concerning our manuscript entitled “Remaining useful life prediction of gear pump based on deep sparse autoencoders and multilayer bidirectional long short-term memory network” (ID: processes-1984081). Those comments are all valuable and very helpful for revising and improving our paper, as well as the important guiding significance to our research. We have studied the comments carefully and have made corrections which we hope meet with approval. Revised portions are marked by the “Track Changes” function in the paper. The main corrections in the paper and the responses to the reviewer’s comments are as flowing:
Comment_1: To avoid the modeling and simulation of complex systems without consulting a high-level design engineer.
Response: We are confused by your reference to "simulation", as the paper does not use any simulation of complex systems. The tests were performed on four real gear pumps and the vibration data used for the analysis was obtained by them.
Comment_2: To avoid the "chains" of algorithms without individual validation using widely accepted models.
Response: It is true as the reviewer suggested that such "chains" do exist in the paper, but they are helpful in describing our algorithm. We respect your suggestion and have removed some of the descriptions in the theoretical section, but the important points are still retained.
Comment_3: To study the new types of sensors without contact which can give precious information concerning each component of the studied assembly.
Response: As the reviewer suggested that the acquisition of vibration signals is contact, and the acquisition of sound signals is non-contact. Our team is also working on fault diagnosis techniques based on noise signals, and we will try to use noise signals as analysis objects in the future. We will give an outlook on this at the end of the article.
Special thanks to you for your good comments.
We tried our best to improve the manuscript and made some changes in the manuscript. These changes will not influence the content and framework of the paper. And here we did not list the changes but marked them by the “Track Changes” function in the revised paper.
We appreciate for your warm work and hope that the correction will meet with approval.
Once again, thank you very much for your comments and suggestions.
Sincerely yours,
Pei-yao Zhang, PhD

Reviewer 3 Report
This is a good paper, with a very good support and mathematical approach. The paper contains numerous references to other good papers in the field.
However, I personally doubt its utility in the industry. Pumps can break down very easily or work without intervention for very long periods of time. Companies often adopt preventive and corrective maintenance programs for their hydraulic installations. I think that the authors should have been written about maintenance in this paper.
The algorithm is interesting and well supported mathematically from a theoretical point of view, it would be possible to write it in the form of a program in Python, C, etc. to be more easily optimized and improved?
The test presented involved real pumps and installation or it is a simulation ?
I think that a test on real pumps should be considered, if not now, then in the near future.
More details could be presented about the LabVIEW program.
Overall, the paper has a good impact in the field.

Author Response
Dear Reviewers:
Thank you for your letter and for the reviewers’ comments concerning our manuscript entitled “Remaining useful life prediction of gear pump based on deep sparse autoencoders and multilayer bidirectional long short-term memory network” (ID: processes-1984081). Those comments are all valuable and very helpful for revising and improving our paper, as well as the important guiding significance to our research. We have studied the comments carefully and have made corrections which we hope meet with approval. Revised portions are marked by the “Track Changes” function in the paper. The main corrections in the paper and the responses to the reviewer’s comments are as flowing:
Comment_1: I doubt its utility in the industry. Pumps can break down very easily or work without intervention for very long periods. Companies often adopt preventive and corrective maintenance programs for their hydraulic installations. We think that the authors should have written about maintenance in this paper.
Response: Indeed, as the reviewer said that, RUL technology is still in the research stage in hydraulic equipment and is not mature. So, some of the descriptions in the article may not be the same as you think. If maintenance of the equipment is to be achieved, research on the fault diagnosis is also needed. When the health status indicator of the equipment falls below a threshold, a fault diagnosis program is initiated to locate possible faults and provide guidance for the maintenance of the equipment. In addition, we have also researched fault diagnosis techniques, but this paper focuses on the RUL technique for the equipment and therefore does not present it in a relevant way.
Comment_2: The algorithm is interesting and well supported mathematically from a theoretical point of view, it would be possible to write it in the form of a program in Python, C, etc. to be more easily optimized and improved?
Response: As you said that write program in Python, C, etc. will be easily optimized and improved. And our algorithm described in this article is written in Python, exactly.
Comment_3: The test presented involved real pumps and installation or it is a simulation?
Response: I'm sorry to have troubled you. The tests were run on real pumps, and the physical information was not obtained through simulation software such as AMESim. Moreover, the simulation of the vibration signal is difficult.
Comment_4: More details could be presented about the LabVIEW program.
Response: Over the LabVIEW program used for the experiments, we have added a back panel of the program in section 3.1, which you can use to learn more information about the program.
I also saw the PDF file you uploaded, thank you again for your serious help. We have also revised the article in the appropriate place for the issues you mentioned in the PDF file.
Special thanks to you for your good comments.
We tried our best to improve the manuscript and made some changes in the manuscript. These changes will not influence the content and framework of the paper. And here we did not list the changes but marked them by the “Track Changes” function in the revised paper.
We appreciate for your warm work and hope that the correction will meet with approval.
Once again, thank you very much for your comments and suggestions.
Sincerely yours,
Pei-yao Zhang, PhD

Reviewer 4 Report
The Research Paper needs the following revisions and is subject for re-review, and after re-review, the final decision for the paper will be done:
1. Add in the last lines of abstract, in what %age and in what parameters the proposed work is better as compared to existing techniques, and what is the overall analysis.
2. Add more information to the introduction section with regard to the Background, scope and problem Definition. Add Objectives of the paper at end of Introduction and add Organization of paper.
3. Literature review is missing in the paper. Add min 10-15 papers and proper explain with regard to the proposed work, novelty and results, and even add 9-15 lines at end of Literature review, that shows the research gaps which led to the design of proposed methodology.
4. Add details of Proposed methodology with regard to Theory, System Model, Algorithm and Flowchart.
5. Add some Real time case study based discussion to the paper.
6. Add future scope to the paper.
Author Response
Dear Reviewers:
Thank you for your letter and for the reviewers’ comments concerning our manuscript entitled “Remaining useful life prediction of gear pump based on deep sparse autoencoders and multilayer bidirectional long short-term memory network” (ID: processes-1984081). Those comments are all valuable and very helpful for revising and improving our paper, as well as the important guiding significance to our research. We have studied the comments carefully and have made corrections which we hope meet with approval. Revised portions are marked by the “Track Changes” function in the paper. The main corrections in the paper and the responses to the reviewer’s comments are as flowing:
Comment_1: Add in the last lines of the abstract, in what %age and in what parameters the proposed work is better as compared to existing techniques, and what is the overall analysis.
Response: We are very sorry for our negligence. We have added a quantitative description and an analytical summary in the abstract section.
Comment_2: Add more information to the introduction section about the background, scope, and problem definition. Add the Objectives of the paper at end of the Introduction and add the organization of the paper.
Response: We have made addition according to the Reviewer’s comments. The background, scope, definition, and purpose related to this study have been added in the Introduction section. And added a structure chart of the paper at the end of this section.
Comment_3: Literature review is missing in the paper. Add min 10-15 papers and properly explain about the proposed work, novelty and results, and even add 9-15 lines at end of the Literature review, that show the research gaps which led to the design of the proposed methodology.
Response: As the Reviewer suggested, we have added a literature review in the Introduction section to organize and briefly analyze the literature on feature extraction, feature fusion, RUL prediction, etc. At the end of the literature review, the problems of existing research are analyzed and the research objectives of this paper are clarified.
Comment_4: Add details of the proposed methodology about Theory, System Model, Algorithm and Flowchart.
Response: As the Reviewer suggested, we have added several illustrative pictures and the experiment’s flow diagram with related explanatory content in the second and third sections. They could provide a more detailed description of the proposed methodology and the work done in this paper.
Comment_5: Add some Real time case study based discussion to the paper.
Response: We have made addition according to the Reviewer’s comments. In the literature review, an article on real-time fault diagnosis is described. In addition, real-time analysis is discussed as an outlook at the end of the article.
Comment_6: Add future scope to the paper.
Response: We have made addition according to the Reviewer’s comments. At the end of the article, we have added a look into the future. The contents include the feasibility of real-time health status assessment of hydraulic pumps; the feasibility of using non-contact-time acquisition signals such as noise signals as analysis data; and the integrated application of RUL prediction and fault diagnosis techniques.
Special thanks to you for your good comments.
We tried our best to improve the manuscript and made some changes in the manuscript. These changes will not influence the content and framework of the paper. And here we did not list the changes but marked them by the “Track Changes” function in the revised paper.
We appreciate for your warm work and hope that the correction will meet with approval.
Once again, thank you very much for your comments and suggestions.
Sincerely yours,
Pei-yao Zhang, PhD
